# Self-Reported and Device-Measured Physical Activity in Leisure Time and at Work and Associations with Cardiovascular Events—A Prospective Study of the Physical Activity Paradox

**DOI:** 10.3390/ijerph182212214

**Published:** 2021-11-20

**Authors:** Roman P. Kuster, Philip von Rosen, Wilhelmus J. A. Grooten, Ing-Mari Dohrn, Maria Hagströmer

**Affiliations:** 1Division of Physiotherapy, Department of Neurobiology, Care Sciences and Society, Karolinska Institutet, 141 83 Stockholm, Sweden; roman.kuster@hotmail.ch (R.P.K.); philip.von.rosen@ki.se (P.v.R.); wim.grooten@ki.se (W.J.A.G.); ing-mari.dohrn@ki.se (I.-M.D.); 2IMES Institute of Mechanical Systems, School of Engineering, ZHAW Zurich University of Applied Sciences, 8400 Winterthur, Switzerland; 3Women’s Health and Allied Health Professionals Theme, Medical Unit Occupational Therapy and Physiotherapy, Karolinska University Hospital, 171 77 Stockholm, Sweden; 4Academic Primary Health Care Center, Region Stockholm, 104 31 Stockholm, Sweden

**Keywords:** attitude, behavior, and change study, cardiovascular disease, cox proportional hazard ratio, leisure time physical activity, occupational physical activity, prospective cohort study

## Abstract

The beneficial health effects of physical activity, in particular moderate-to-vigorous physical activity (MVPA), are well documented, but there is an ongoing scientific debate whether the domain matters, i.e., whether leisure time physical activity is beneficial and occupational physical activity is detrimental to health, referred to as the physical activity paradox. The present study, therefore, analyzed the association between self-reported and device-measured physical activity and cardiovascular events in both domains. A representative sample of 807 individuals was followed for 14.6 ± 1.1 years, in which 59 cardiovascular events occurred. For self-reported data, Cox proportional hazard models showed no effect of physical activity in leisure and at work, while for device-measured MVPA, beneficial associations with total time spent in MVPA and occupational time spent in MVPA were found, but not for leisure time spent in MVPA. When accounting for both domains in the same model, the associations disappeared. These results indicate that it matters how physical activity is measured and that MVPA is beneficial for cardiovascular health, but the domain in which MVPA occurs does not seem to matter.

## 1. Introduction

The beneficial effect of physical activity on health is well documented [1,2], reflected by the large and increasing number of national and international physical activity guidelines [3,4,5]. The evidence for positive health effects is particularly strong for moderate-to-vigorous physical activity (MVPA) and, according to the current evidence, not affected by the domain in which the activity occurs [6]. The positive health effects of physical activity mainly derive from studies using self-reported leisure time MVPA, while there is research suggesting that occupational physical activity does not have the same positive effects [7], often referred to as the physical activity paradox [8]. For example, a decreased risk for long-term sickness absence among workers with moderate to high leisure time physical activity but an opposite effect for occupational physical activity; moderate to high occupational physical activity increased the risk for long-term sickness absence has been observed [8]. Similar findings have been reported for cardiovascular disease risks [9,10].

A recent umbrella review synthesized the health effects of occupational and leisure time physical activity [11] and found either no or a positive effect on the investigated health outcomes in both domains, but no negative effects. The positive effects differed however in the magnitude for coronary heart diseases, diabetes type 2, and distal colon cancer, with leisure time physical activity having a larger beneficial effect than occupational physical activity [11]. This might be explained that physical activity at work and in leisure can be different in nature and degree of self-control [12]. In addition, the conflicting results from studies investigating the associations between occupational physical activity and cardiovascular health mainly derive from self-reported physical activity, e.g., [13]. The limitations of self-reported physical activity measurements are well known, and include, among others, social desirability, recall bias, and the inability to assess the intensity of physical activity, leading to low-to-moderate associations between self-reported and device-measured physical activity, both at leisure and at work [14,15,16,17]. However, prospective studies assessing the physical activity paradox with device-based methods are rare [18]. We hypothesized that using the same concept, device-measured MVPA, in both domains will have a similar effect on cardiovascular events, while the results from self-reported MVPA may differ.

This study aimed to analyze the effect of occupational and leisure time physical activity on cardiovascular events in a population-based cohort using both self-reported and device-measured physical activity.

## 2. Materials and Methods

This prospective cohort study used data from the Attitude, Behavior, and Change Study (ABC) study collected from September 2000 to December 2001. Participant recruiting and recording are described in detail elsewhere [19]. In short, a randomly selected sample of 3300 adults 18–75 years was drawn from the Swedish population register, 2265 were reached by phone, and 1556 agreeing to participate received an accelerometer (ActiGraph MT1 model 7164, Manufacturing Technology Inc., Fort Walton Beach, FL, USA) and a questionnaire by mail. The selected population was representative of the Swedish population [19]. The accelerometer was initialized as recommended by the manufacturer and recorded the counts-per-minute (cpm) on the vertical sensor axis in 1-min epochs. Participants were instructed to wear the accelerometer on a belt around the lower back during waking hours for 7 days, only removing it for water-based activities, and send it back by mail together with a questionnaire. Valid physical activity data were provided from 1220 participants. The analytical sample included (self-)employed participants, (excluding students and retired persons) without previous heart disease, leading to a sample size of *n* = 807.

### 2.1. Outcome

Participants were followed prospectively from the first day of accelerometer measurement until the occurrence of a fatal or non-fatal cardiovascular event or censoring by death from other causes or on 31 December 2015. Information on death for the years 2002 to 2015 was obtained in 2016 from the National Board of Health and Welfare’s Cause of Death Register, information on the first diagnosis of cardiovascular disease for the years 2002 to 2015 was obtained in 2016 from the National Patient Register in Sweden. Cardiovascular diseases, including stroke, were defined as I10-I15, I20-I25, I60-I79 according to the International Classification of Disease (ICD-10), excluding subarachnoid hemorrhage and essential hypertension.

### 2.2. Self-Reported Physical Activity

Self-reported leisure time physical activity was determined using the question: How much exercise do you get in your leisure time? Participants had five possible answers to choose from: (1) no exercise at all, (2) exercise every now and then, (3) exercise about once a week, (4) exercise regularly about twice a week, and (5) exercise regularly quite vigorously at least twice a week. The answer was dichotomized into: those exercising less than twice a week (answer 1 to 3) to the sedentary group, and those exercising at least twice a week (answer 4 and 5) to the active group.

Self-reported occupational physical activity was derived from the job title reported by the participants [20,21]. The occupations were coded into 40 categories using the US Census Bureau’s 2000 Indexes of Industry and Occupations [22]. Each category was then assigned to one of three occupational physical activity groups: (1) sedentary group, including office-based jobs like executives, administrators, secretaries, distributing clerks, and records-processing occupations; (2) mixed group, including jobs that could not clearly be assigned to one of the other two groups; (3) active group, including jobs like waiters/waitresses, cleaning and building services, construction workers, and manual workers.

### 2.3. Deviced-Measured Physical Activity

Device-measured leisure and occupational physical activity were determined from the cpm files of the accelerometer and processed with the statistical software R (v4.0.5). First, periods of non-wear (60 consecutive minutes of 0 cpm with an allowance of up to 2 min with 100 cpm at most [23]) were excluded from the analysis. Second, only valid days were kept in the analysis. A valid day was defined as having at least 10 recording hours and no counts above 20,000 cpm, indicating sensor malfunction [19]. To determine a proxy for the occupational time we assumed the majority was working daytime. Only participants with at least 1 weekday were kept in the analysis and a weekday was only registered if at least 6 h were recorded between 08:00 and 16:00. Thus, the majority of individuals working at night are likely excluded since they do not have 6 h of wear time recorded between 08:00–16:00. The chosen daily time span is supported by regular working hours in Sweden at that time and has also been shown to correspond to working hours in another study [24]. The remaining wear time was classified as leisure time.

Each minute was subsequently categorized into sedentary behavior, light-intensity physical activity, and MVPA using the cut-point of 100 cpm [23] and 2020 cpm [25]. The activity classification was then aggregated over all valid days for each participant and expressed relative to the recording time, separately for total time, leisure time, and occupational time. The relative time spent in MVPA in total, during leisure time, and during occupational time was subsequently used to analyze the associations with cardiovascular events.

### 2.4. Covariates

The questionnaire provided information on age, sex, smoking status (never/former or current), body mass index (BMI) calculated from height and weight, previous diseases (hypertension, heart disease, cancer, diabetes, arthritis), and education (university degree/no university degree). These were used to adjust for possible confounders to cardiovascular events.

### 2.5. Data Analyses

The associations between total time, leisure time, or occupational physical activity and cardiovascular events were investigated with Cox proportional hazard models to estimate hazard ratios (HR) with 95% confidence intervals (CI). All models used the follow-up time as the underlining time scale and were adjusted for known confounders in the ABC study data: age, sex, education, and previous diseases [19]. For self-reported physical activity, Model A investigated the leisure time physical activity group (sedentary, active), Model B the occupational physical activity group (sedentary, mixed, active), and Model C the combined effect of leisure and occupational physical activity by including both variables. Interaction effect between leisure and occupational physical activity was tested in Model C and included if significant. Nine participants had missing data for leisure time physical activity and were, thus, excluded from those analyses (including 1 event). For device-measured physical activity, tertiles of total MVPA (least, medium, and most active) were used. Separate models investigated the effect of time spent in MVPA regardless of the domain (Model I), leisure time spent in MVPA (Model II), occupational time spent in MVPA (Model III), and the combined effect of MVPA during leisure and occupational time (Model IV) by the mutual adjustment of leisure and occupational MVPA. All models were checked for the proportional hazard assumption, influential observations, and linearity between the log hazard and independent variables. *p*-values ≤0.05 were considered statistically significant. Sensitivity analyses were conducted by excluding individuals younger than 35 years at baseline, and by excluding individuals with less than four days of accelerometer recording or no weekend day (Appendix A, Table A1). Data analysis was conducted in the statistical software R (v3.5.1).

## 3. Results

During a mean follow-up time of 5397 ± 410 days (equal to 14.6 ± 1.1 years) a total of 59 first cardiovascular events occurred, including three deaths. Table 1 illustrates the characteristics of the study population. At baseline, participants were between 19 and 71 years, with an average age of 43 years. A majority, 66%, had no university degree, and 88% reported no history of previous diseases. Most participants (54%) provided seven valid days of device-measured data, and 89% provided data from at least four days, including at least one weekend day. The average total wear time was 14.7 ± 1.2 h per day. This resulted in an average recording time per person of 90 ± 22 h, of which 61% was collected during leisure time, and 39% was collected during occupational time. Participants spent a similar percentage of time in MVPA in both domains.

Self-reported leisure and occupational physical activity were not associated with cardiovascular events (Table 2), neither was their interaction.

Device-measured total and occupational physical activity were associated (*p* < 0.05), with cardiovascular events when comparing the most active with the least active tertiles, but not leisure time physical activity (Table 3). When accounting for occupational and leisure time physical activity in the same model (Model IV), none of the domains were associated with cardiovascular events. Similar results were found in the sensitivity analysis (Appendix A, Table A1).

## 4. Discussion

The present study is among the first to study the physical activity paradox both with self-reported and device-measured physical activity. Self-reported physical activity did not show significant associations with cardiovascular events. However, the non-significant point estimates are indicating towards the physical activity paradox, with a beneficial association for leisure time physical activity and a detrimental association for occupational time physical activity. For the device-measured physical activity, we found, contrary to the physical activity paradox, that total MVPA, as well as MVPA, at work were beneficially associated with cardiovascular events, while MVPA during leisure time showed a beneficial but non-significant association with cardiovascular events. When accounting for both domains in the same model the beneficial association from MVPA at work disappeared. This indicates that the domain in which MVPA occurs does not matter. In addition, one might compensate for a low level of MVPA in one domain with a high level of MVPA in the other domain. From a physiological view, it is reasonable to think that the health benefits from aerobic MVPA would be the same regardless of where the activity is performed. Thus, the detrimental effects of occupational MVPA previously shown [11], based on self-report data, might be influenced by other factors related to physical or mental load at work rather than device-measured MVPA, which involves more aerobic elements. Our results also indicate that the association between domain-specific physical activity and cardiovascular events depends on how physical activity is measured. Self-reported and device-measured physical activity have previously shown only a low-to-moderate association, which might explain this result [15,16,17].

The different nature of physical activity at work and in leisure is undisputed. For example, physical activity at work might be characterized by low intensity, long-lasting static activities performed with low self-control, while physical activity during leisure time, might be more intermittent, dynamic, and include more MVPA, with a high degree of self-control [12]. Even though it has been shown that the total volume of MVPA contributes to the greatest health benefits [1,2,26], the accumulated time in lower intensity activities is also beneficial [27]. However, light-intensity physical activity is almost impossible to accurately assess with questionnaires [28]. Another major challenge with the use of questionnaires is that physical activity at work and during leisure might be differently perceived. For example, jobs like waiters/waitresses and construction workers could all be considered and perceived as active, but the intensity, duration, and frequency of physical activity might be very different. This highlights the importance of using device-based methods, as in this study, to explore potential domain-dependency health effects of physical activity.

Measuring physical activity during work, irrespective of using self-reports or device-based measures is a challenge. Accelerometry captures MVPA mainly deriving from ambulatory movement at leisure and work, but not all exercise activities or every single work task, such as upper body movements, or physical load, such as lifting. Still, device-measured total daily MVPA is associated with cardiorespiratory fitness as well as cardiovascular disease and mortality [27]. An important difference to other studies investigating the physical activity paradox with self-reports is the classification of occupational physical activity based on job type [20,21], while most other studies ask their participants about their occupational physical activity level. We do not know how this might have affected our results compared to others. Yet, our results are in line with a recent study from the UK Biobank [29]. Within the working population, they found no evidence of differences in mortality by occupational physical activity group when comparing those reporting higher levels to the lowest reference group. Until more evidence on the potential beneficial effects on cardiovascular health from device-measured occupational physical activity is provided, physical activity during leisure time should be facilitated and promoted to individuals irrespective of occupation. The pandemic has further highlighted the role of socioeconomic and environmental influences on physical activity behaviors. Moreover, the boundaries between leisure time and occupational time have, in many occupations (e.g., office workers), been increasingly blurred by the pandemic, making it difficult to study the physical activity paradox in the future.

An important strength of this study is the use of both device-measured and self-reported physical activity to study the physical activity paradox. With the device-measured physical activity, we used the time spent in MVPA, the most important physical activity behavior contributing to health. Other important strengths of this study are the long follow-up time and the study sample with participants drawn from the Swedish population register [19,27]. The outcome measure of our study, cardiovascular events, was based on highly reliable registry data, which must be considered another strength of our analysis. The Swedish cause of death register is a largely complete and high-quality data source [30], and the National Patient Register includes the main diagnosis for 99% of all hospital in-patient admissions and 96% of all outpatient visits, with high validity [31].

We acknowledge the current discussion about whether the physical activity paradox might be explained by the inclusion of the right combination of confounders [13,32,33]. To limit the number of covariates in the models, we tested a-priory selected confounders based on previous studies [1,34] and only adjusted our models for confounders statistically associated with the outcome [35]. Included confounders were self-reported, and as in any observational study, our results may be subject to residual confoundings, such as excessive alcohol consumption or family history of cardiovascular disease.

Data on physical activity and covariates were only collected at baseline, and it remains unknown whether participants would have answered the questions regarding exercise habits and job type differently if asked repeatedly. However, we can assume that there is relative stability in work-related physical activity over time, since not many change their occupation according to a Swedish report [36]. For the device-measured physical activity, we know from a repeated measurement on a sub-cohort of the ABC sample that there were no significant changes for average intensity and total time in MVPA over the first six years after the baseline measurements [37]. Additionally, our total physical activity data are very similar to a more recent population-based Swedish study [38].

The separation into the two domains used the day of the week and the time of the day (i.e., the time between 08:00 and 16:00 on weekdays was considered occupational time, all other recorded time was considered leisure), which was supported by previous studies [24]. Individuals working the night shift were most likely excluded from the analysis leaving the remaining shift workers as daytime workers in the analysis. This proxy of using the time of the day as a proxy for occupational time could have been a limitation in our study. In the Swedish working population, about 20% is working shift [39]. There is some evidence that shift workers have a higher risk for adverse health events, and since shift work occurs more frequently in the group with high occupational physical activity, there could have been a differential misclassification of physical activity [39,40]. This could have led to a bias of the effects in either under- or overestimation of the risk estimates. Still, only a minor part of the shift workers’ working time was misclassified and we believe that the potential differential misclassification has led to an underestimation of the risk estimates, i.e., a smaller dilution of our results. Future studies should use an individual classification of working times.

Even though the ABC cohort was a nationally representative sample, as in any research study, the participants may be healthier and more physically active than the general population which may affect the number of events. Still, the amount of first cardiovascular events (7.3%) in our population is similar to other studies [13], but the sample size and number of events may have limited our statistical power. Consequently, we have less power in the models with the most parameters (e.g., Model C, Model IV), explaining the wide confidence intervals associated with their point estimates. Future studies using device-based assessment of physical activity are needed to confirm our findings. To distinguish the domains in which the activity occurs, we recommend that future studies will use a complimentary diary that could enable cross-validation of the data sources. Future research should also investigate the possible differences between measured and perceived intensity and the amount of physical activity performed at work, since it is suggested that many occupations may not be as active as reported [29]. Even though we used both self-reports and movement devices, we are aware that these two methods assess physical activity in different ways, associated with various limitations, that affect our findings. Last, we consider it important to mention that the present study is observational, and reverse causation is an issue that remains unknown.

## 5. Conclusions

This study showed that the domain-dependent association of physical activity and cardiovascular health differs in relation to how physical activity is measured. The study also showed a beneficial association of device-measured total MVPA and MVPA at work with the occurrence of cardiovascular events. However, the association of MVPA at work disappeared when accounting for MVPA during leisure time, indicating that the domain in which MVPA occurs might not matter.

## Figures and Tables

**Table 1 ijerph-18-12214-t001:** Baseline characteristics of the full study sample and by tertiles of device-measured moderate-to-vigorous physical activity.

	All	Tertiles of Total MVPA		
		Least Active	Medium Active	Most Active
Sample size	807 (100)	269 (33.3)	269 (33.3)	269 (33.3)
Men	365 (45.2)	102 (37.9)	122 (45.4)	141 (52.4)
Age, years	42.8 ± 11.5	45.5 ± 11.1	41.8 ± 11.2	41.0 ± 11.7
BMI, kg/m^2^	25.0 ± 3.6	25.8 ± 4.3	24.7 ± 3.1	24.5 ± 3.2
Smoker	186 (23.0)	76 (28.3)	55 (20.4)	55 (20.4)
University degree	278 (34.4)	70 (26.0)	103 (38.3)	105 (39.0)
Previous diseases *	97 (12.0)	42 (15.6)	28 (10.4)	27 (10.0)
**Self-reported PA**				
Occupational PA				
sedentary	323 (40.0)	103 (38.3)	114 (42.4)	106 (39.4)
mixed	407 (50.4)	143 (53.2)	134 (49.8)	130 (48.3)
active	77 (9.5)	23 (8.6)	21 (7.8)	33 (12.3)
Leisure time PA ^#^				
sedentary	519 (64.3)	208 (77.3)	166 (61.7)	145 (53.9)
active	279 (34.6)	58 (21.6)	99 (36.8)	122 (45.4)
**Device-measured PA**				
Total wear time, h/day	14.7 ± 1.2	14.7 ± 1.3	14.9 ± 1.1	14.6 ± 1.2
MVPA (%)	4.1 ± 2.6	1.6 ± 0.7	3.6 ± 0.6	7.0 ± 2.2
light-intensity PA (%)	39.8 ± 9.9	38.5 ± 10.9	39.7 ± 9.2	41.2 ± 9.2
sedentary (%)	56.1 ± 10.6	59.8 ± 11.1	56.7 ± 9.2	51.8 ± 9.8
Leisure time, h/day	9.0 ± 1.5	8.9 ± 1.6	9.2 ± 1.4	8.9 ± 1.6
MVPA (%)	4.1 ± 2.9	1.6 ± 0.9	3.7 ± 1.1	7.0 ± 3.0
light-intensity PA (%)	38.7 ± 9.1	37.1 ± 10.2	39.1 ± 8.5	39.8 ± 8.4
sedentary (%)	57.2 ± 10.0	61.3 ± 10.5	57.2 ± 8.6	53.2 ± 9.0
Occupational time, h/day	7.8 ± 0.3	7.8 ± 0.4	7.8 ± 0.3	7.8 ± 0.3
MVPA (%)	4.1 ± 3.3	1.7 ± 1.2	3.6 ± 1.7	7.0 ± 3.8
light-intensity PA (%)	41.6 ± 14.7	40.7 ± 15.6	40.8 ± 14.6	43.3 ± 13.8
sedentary (%)	54.3 ± 15.7	57.6 ± 15.8	55.6 ± 15.0	49.7 ± 15.2

Data presented as numbers (percentages) or mean ± standard deviation. Abbreviations: BMI = body mass index, MVPA = moderate-to-vigorous physical activity, PA = physical activity. * self-reported history of hypertension, heart disease, cancer, diabetes, arthritis. ^#^ missing data for 9 participants.

**Table 2 ijerph-18-12214-t002:** Cox proportional hazards for self-reported physical activity, *n* = 798 and 58 events (due to missing self-reported leisure time physical activity).

	Leisure PA		Occupational PA		
	Sedentary (Ref)	Active	Sedentary (Ref)	Mixed	Active
Events, *n* (%)	42 (8.1)	16 (5.7)	23 (7.1)	27 (6.6)	9 (11.7)
	HR (95% CI)	HR (95% CI)	HR (95% CI)	HR (95% CI)	HR (95% CI)
Model A	1.0	0.65 (0.36–1.16)	-	-	-
Model B	-	-	1.0	0.94 (0.52–1.71)	1.15 (0.50–2.63)
Model C	1.0	0.74 (0.41–1.32)	1.0	1.05 (0.59–1.88)	1.65 (0.73–3.74)

Model A includes leisure PA, Model B includes occupational PA, Model C includes leisure and occupational PA. All models are adjusted for age, sex, education and previous disease. Abbreviations: CI = confidence interval, HR = hazard ratio, PA = physical activity.

**Table 3 ijerph-18-12214-t003:** Cox proportional hazards for device-measured moderate-to-vigorous physical activity.

	Tertiles of Total MVPA		
	Least Active (Ref)	Medium Active	Most Active
Events, *n* (%)	26 (9.7)	23 (8.6)	10 (3.7)
	HR (95% CI)	HR (95% CI)	HR (95% CI)
Model I	1.0	1.18 (0.67–2.07)	**0.46 (0.22–0.97)**
Model II (leisure PA)	1.0	0.76 (0.42–1.38)	0.60 (0.31–1.17)
Model III (occupational PA)	1.0	1.31 (0.74–2.30)	**0.43 (0.21–0.92)**
Model IV			
Leisure PA	1.0	0.84 (0.46–1.54)	0.80 (0.38–1.67)
Occupational PA	1.0	1.38 (0.77–2.47)	0.48 (0.21–1.08)

Model I includes wear time PA, Model II includes leisure time PA, Model III includes occupational time PA, Model IV includes leisure time and occupational time PA. All models are adjusted for age, sex, education and previous disease. Significant associations in bold (*p* < 0.05). Abbreviations: CI = confidence interval, HR = hazard ratio, MVPA = moderate-to-vigorous physical activity, PA = physical activity.

## Data Availability

The data presented in this study are available on reasonable request from the last author (M.H.). The data are not publicly available due to ethical restrictions.

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
