# Peer review of "Self-Reported and Device-Measured Physical Activity in Leisure Time and at Work and Associations with Cardiovascular Events—A Prospective Study of the Physical Activity Paradox"

_ijerph, 2021, doi:10.3390/ijerph182212214_

Round 1

Reviewer 1 Report

First, I highlight the excellent work that the authors put into this study. The manuscript investigated the self-reported and device-based physical activity paradox with a large sample of Swedish adults. The brief introduction is well structured, and it contains all the necessary information to understand the study. The "Material and Methods" had detailed information about the study. However, I would recommend highlighting the longitudinal aspects of the study. The "result" section only concentrating on the main result, which has appropriate statistical methods. The authors used several empirical supports in their discussion, and their conclusion is adequate.

Overall, it is a brief but excellent study. I recommend publishing as it is.  

Author Response

The authors thank the reviewer for the positive feed-back

Reviewer 2 Report

Thank you for asking me to review this article. Investigating the effects of leisure and work-related physical activity on cardiovascular events, also in relation to the chosen method of measurement (self-reported or device-based), is important in the current pandemic period. The isolation measures imposed in the fight against COVID-19, and the resulting reduction in opportunities to practice physical activity and sports, have, in fact, led to an increase in sedentary levels. As such, studies that investigate the determinants of the paradox of physical activity through the use of different analytical methodologies can offer useful insights.

The study put forward by the authors is well-described in its method and presents some very interesting results, however, it needs more revision before proceeding to publication.

In this regard, my suggestions are the following:

In the introductory paragraph, the hypotheses that the research intended to test should be expanded upon in order to clarify the context. In particular, with reference to the physical activity performed, in lines 44-48 the authors describe light and/or moderate occupational physical activity as one of the predictors of morbidity, but do not mention what other determinants might adversely affect the onset of cardiovascular disease in this cohort of individuals.

Occupational physical activity, in fact, is often characterised by endurance  and repetitive actions whereas recreational physical activity is usually aerobic, which is more likely to improve fitness and cardiovascular health (doi:10.1136/bmjopen-2016-012692). Furthermore, occupational physical activity, compared to recreational physical activity, is performed with shorter recovery periods and often without adequate control of the working conditions (indoor or outdoor). Blood pressure in this cohort may also increase as a result of continuous exertion such as lifting weights or static postures (typical of occupational physical activity), with consequent adverse health effects. These aspects are briefly described by the authors in the discussion paragraph (lines 198-204) but should be further elaborated in the introductory paragraph (doi: 10.1177/2047487314554866 and doi: 10.5271/sjweh.3476) in order to give more strength to the hypotheses underlying the research question.

Ultimately, the findings are clear and directly derived  from the data analysis, however I suggest that the findings be discussed by clarifying how the research conducted can contribute to the analysis of the pandemic physical activity paradox. For example, given the effects of the COVID-19 pandemic on many aspects of human health, it would be interesting to consider the weight of other factors that now influence recreational physical activity, including, for example, living environments. In fact, given the impossibility for everyone to find space, economic means and time to engage in physical activity in dedicated spaces (i.e. sports facilities and/or gyms) during lockdown, Public Health is considering the possibilities of adapting the built environment also to respond to future emergencies. In relation to the need to consider improving urban spaces for physical activity, several authors address the role of the environment on recreational activities, also suggesting different application models in different territorial contexts (doi: 10.1007/978-3-030-86973-1_18).

Looking more in-depth into these aspects could strengthen the research question by making the study more complete.

Author Response

  1. In the introductory paragraph, the hypotheses that the research intended to test should be expanded upon in order to clarify the context. In particular, with reference to the physical activity performed, in lines 44-48 the authors describe light and/or moderate occupational physical activity as one of the predictors of morbidity, but do not mention what other determinants might adversely affect the onset of cardiovascular disease in this cohort of individuals.

Response: We agree with the reviewer that there are other determinants for onset of cardiovascular disease and premature death. We have taken those into consideration as covariates in the analysis and have justified this in the method section see line 126-127. We have also extended the background to include examples of studies on cardiovascular disease, see line 43-44.

  1. physical activity, in fact, is often characterised by endurance and repetitive actions whereas recreational physical activity is usually aerobic, which is more likely to improve fitness and cardiovascular health (doi:10.1136/bmjopen-2016-012692). Furthermore, occupational physical activity, compared to recreational physical activity, is performed with shorter recovery periods and often without adequate control of the working conditions (indoor or outdoor). Blood pressure in this cohort may also increase as a result of continuous exertion such as lifting weights or static postures (typical of occupational physical activity), with consequent adverse health effects. These aspects are briefly described by the authors in the discussion paragraph (lines 198-204) but should be further elaborated in the introductory paragraph (doi: 10.1177/2047487314554866 and doi: 10.5271/sjweh.3476) in order to give more strength to the hypotheses underlying the research question.

Response: The reviewer is correct that it is likely that physical activity at work and in leisure can for example be different in nature and degree of self-control and this can be the explanation for the paradox. Therefore, it is of interest to test if it is the different nature of physical activity using the same instrument or if it is different concepts used. We have added this justification in the background, see line 59-61.

  1. Ultimately, the findings are clear and directly derived from the data analysis, however I suggest that the findings be discussed by clarifying how the research conducted can contribute to the analysis of the pandemic physical activity paradox. For example, given the effects of the COVID-19 pandemic on many aspects of human health, it would be interesting to consider the weight of other factors that now influence recreational physical activity, including, for example, living environments. In fact, given the impossibility for everyone to find space, economic means and time to engage in physical activity in dedicated spaces (i.e. sports facilities and/or gyms) during lockdown, Public Health is considering the possibilities of adapting the built environment also to respond to future emergencies. In relation to the need to consider improving urban spaces for physical activity, several authors address the role of the environment on recreational activities, also suggesting different application models in different territorial contexts (doi: 10.1007/978-3-030-86973-1_18).

Response: We agree that living environments can influence physical activity both at work and leisure. Our study was conducted prior to the COVID-19 pandemic and without any data on environmental factors so we cannot test such aspects and find it out of focus for this study.

  1. Looking more in-depth into these aspects could strengthen the research question by making the study more complete.

Response: We thank the reviewer for positive words and for highlighting aspects that we agree can strengthen the paper.

Reviewer 3 Report

The paper provides a welcome addition to the literature given the scarcity of objectively measured data linking PA with cardiovascular outcomes. I have some specific queries or comments as follows:

  1. Section 2.3. It is not entirely clear how the occupational time was measured with the accelerometer. Only using the hours between 8am and 4pm seems very restrictive and would thus exclude some active occupation like waitressing? Were data collected for each individual showing whether they were at work or not and their actual times used? Or was it just assumed that "work" happened between the hours of 8am and 4pm and all other times were "leisure"? What about shift workers? This would seem to be a major limitation? Reference 21 refers to "aging" workers with a mean age of 62 years - quite different from this sample of 42 years (range 20-70). It is really an appropriate reference in this regard?
  2. It is perhaps not surprising that self-reported and device measured PA produces different results given the marked differences in how these behaviours are measured! Can they really be compared properly - or does the method outlined in point 1 above have to be used in order to really answer this question?
  3. Was the study sufficiently powered for such analyses? The number of events was fairly low and the overall sample not large.
  4. Rather than one domain compensating for the other (discussion), isn't the lack of effect in terms of overall MVPA more to do with the inability to truly distinguish between work vs leisure time activity with the accelerometer?

Author Response

  1. Section 2.3. It is not entirely clear how the occupational time was measured with the accelerometer. Only using the hours between 8am and 4pm seems very restrictive and would thus exclude some active occupation like waitressing? Were data collected for each individual showing whether they were at work or not and their actual times used? Or was it just assumed that "work" happened between the hours of 8am and 4pm and all other times were "leisure"? What about shift workers? This would seem to be a major limitation? Reference 21 refers to "aging" workers with a mean age of 62 years - quite different from this sample of 42 years (range 20-70). It is really an appropriate reference in this regard?

Response: We agree that the way we measured occupational time can be a limitation. As we did not have data on when they started and stopped work, we selected individuals that reported to work full-time and daytime. We assumed based upon regular working hours in Sweden at that time (2001-2002) a majority is working between 8 am – 4 pm. On top of the national work-time data we supported our choice with a reference and agree that this is not the same age group, but we would argue that in this case the age does should not matter. We are not comparing amount of PA, just the approximate time the start and end work. See method section line 112-114.

The fact that we have not included shift workers is another limitation, that we discuss in the discussion. Those working nighttime would have been excluded from the analysis as we as inclusion had a least 6 hours wear time between 8-4. Please find this discussed on line 264-269.
The separation into the two domains used the day of the week and the time of the day (i.e. time between 08:00 and 16:00 on weekdays was considered occupational time, all other recorded time was considered leisure), which was supported by previous studies [21]. Since shift work occurs more frequent in the group with high occupational physical activity [37], this could have led to a differential misclassification of physical activity, leading to a dilution of the effects. Future studies should try use an individual classification of working times.

  1. It is perhaps not surprising that self-reported and device measured PA produces different results given the marked differences in how these behaviours are measured! Can they really be compared properly - or does the method outlined in point 1 above have to be used in order to really answer this question?

Response: We agree that self-reported and device measured physical activity does not measure the same thing. We were not interested to compare in that way more to test if MVPA measured with different instruments (i.e capturing different concepts, but the same concept in both domains) yields the same results. This goes back to our “hypothesis” that it might not be a paradox when considering the same aerobic MVPA as measured using for example a device. It is probably something else as discussed i.e repetitive, static PA with low self-control etc. This also highlight how difficult it is to study the PA-paradox.

  1. Was the study sufficiently powered for such analyses? The number of events was fairly low and the overall sample not large.

Response: We agree that the number of events is low and that this is a limitation with the study. Our results indicate that it does not matter in which domain MVPA occurs. Still, the amount of first cardiovascular events (7,3%) in our population is similar to other studies (Holtermann, 2021), but the few events may have limited our statistical power, yet the long follow-up time (15 years) is a strength. Further studies using device-based assessment of physical activity are needed to confirm our findings. See the discussion line 272-275.

  1. Rather than one domain compensating for the other (discussion), isn't the lack of effect in terms of overall MVPA more to do with the inability to truly distinguish between work vs leisure time activity with the accelerometer?

Response: We agree that we should elaborate more on this in the discussion. We found an effect of total time in MVPA, when split by domain we find a similar trend but not significant. This can be a power problem as well as highlighting how difficult it is to measure MVPA using accelerometry in different domains. The accelerometer measured MVPA, i.e using the same measure for MVPA in both domains seem to yield the same positive effect. This goes back to our “hypothesis” that it might not be a paradox when considering the same aerobic MVPA. It is then probably something else as discussed i.e., repetitive, static PA with low self-control etc. We have now elaborated more on this in the discussion see line 201-206.

Round 2

Reviewer 2 Report

The authors responded to each of the suggestions highlighted with the exception of the final comment, in which the authors describe as "out of focus" a possible consideration on the pandemic context in progress. It is clear that the study presented relates to the pre-pandemic period, but nevertheless, in my opinion it is impossible not to consider whether and how this study can contribute to the not so much experiential as theoretical improvement of the physical activity paradox (through considerations or food for thought) also in light of the determinants that the pandemic has highlighted about these aspects. Updating a study by considering it and deepening it also through different interpretations can only offer added value to scientific research. I therefore suggest that the authors discuss how the research conducted can contribute to the analysis of the physical activity paradox even in the light of the pandemic era. In fact, given the impossibility for everyone to find space, economic means and time to devote themselves to physical activity in dedicated spaces during the lockdown, the Public Health is evaluating the possibility of adapting the built environment to respond to future emergencies. In my opinion, it would be useful to include a brief comment on this aspect.

Author Response

Response: We have now added a comment on how the study in the light of the pandemic era can contribute. See lines 243-250

Still, until more evidence on the potential beneficial effects on cardiovascular health from device-measured occupational physical activity is provided, physical activity during leisure-time should be facilitated and promoted to individuals irrespective of occupation. The pandemic has further highlighted the role of socioeconomic and environmental influences on physical activity behaviors. Moreover, the boundaries between leisure time and occupational time have, in many occupations (e.g., office workers), been increasingly blurred by the pandemic, making it difficult to study the physical activity paradox in the future.

Reviewer 3 Report

The authors have responded to each query outlined, but in my view, rather superficially. Each of these points needs to be incorporated and discussed in more detail in the manuscript itself. There are considerable limitations in the comparisons undertaken such that these should be very clearly stated. Please re-submit with a more complete response and appropriate changes to the manuscript.

Author Response

Response: We understand that our response and changes was not sufficient and have now further described and discussed the points raised, in specific:

  1. Occupational time:

We agree that the way we measured occupational time can be a limitation. As we did not have data on when they started and stopped work, we assumed that a majority was working daytime, based upon Statistic Sweden data (2001-2002) and the job categories used for the self-reported classification. We assumed based upon regular working hours in Sweden at that time (2001-2002) a majority is working between 8 am – 4 pm. On top of the national work-time data we supported our choice with a reference and agree that this is not the same age group, but we would argue that in this case the age does should not matter. We are not comparing amount of PA, just the approximate time the start and end work.

The fact that shift workers are included as daytime workers is another limitation. Those working nighttime would have been excluded from the analysis as we as inclusion had a least 6 hours wear time between 8-4. Still those working for example in the evenings and weekends might lead to misclassification.

We have in more detail described how the occupational time was calculated and the assumptions made for this, see section 2.3
To determine a proxy for occupational time we assumed the majority was working daytime. Only participants with at least 1 weekday were kept in the analysis and a weekday was only registered if at least 6 hours were recorded in between 08:00 and 16:00., weekdays had to contain at least 6 hours recorded in between 08:00 and 16:00. Thus, the majority of individuals working at night are likely excluded since they do not have 6 hours of wear time recorded between 08:00-16:00. The chosen daily time span was defined is supported by regular working hours in Sweden at that time and has also shown to correspond to working hours in another study occupational time [24]., and only participants with at least 1 weekday were kept in the analysis. The remaining wear time was classified as leisure time.

We have also extended the discussion in this manner, see line 269-280
The separation into the two domains used the day of the week and the time of the day (i.e. time between 08:00 and 16:00 on weekdays was considered occupational time, all other recorded time was considered leisure), which was supported by previous studies [24]. Individuals working night shift were most likely excluded from the analy-sis leaving the remaining shift workers as daytime workers in the analysis. This proxy of using the time of the day as a proxy for occupational time could have been a limitation in our study. In the Swedish working population about 20% is working shift [39]. There is some evidence that shift workers has a higher risk for adverse health events, and since shift work occurs more frequent in the group with high occupational physical activity, there could have been a differential misclassification of physical activity [39, 40]. This could have led to bias of the effects in either under- or overestimation of the risk estimates. Still, only a minor part of the shift workers’ working time was misclassified and we believe that the potential differential misclassification has led to an underestimation of the risk estimates, i.e. a smaller dilution of our results. Future studies should try use an individual classification of working times.

  1. Regarding self-reported and objectively measured occupational physical activity:

    We agree that self-reported and device measured physical activity does not measure the same thing. Even though we used both self-reports and movement devices, our study was not primary designed to assess these associations. To distinguish the domains in which the activity occurs, future studies should use a complementary diary that could enable a cross-validation of the data sources. Future research should also investigate the possible differences between measured and perceived intensity and amount of physical activity performed at work. This also highlight how difficult it is to study the PA-paradox.

    We have extended the discussion, which also relates to the comment below on domains see lines 300-307.

To distinguish the domains in which the activity occurs, we recommend that future studies will use a complementary diary that could enable a cross-validation of the data sources. Future research should also investigate the possible differences between measured and perceived intensity and amount of physical activity performed at work, since it is suggested that many occupations may not be as active as reported [29]. Even though we used both self-reports and movement devices, we are aware that these two methods assess physical activity in different ways, associated with various limitations, that affect our findings.

  1. Regarding power:
    We agree that the number of events is low and that this is a limitation with the study. Our results indicate that it does not matter in which domain MVPA occurs. Still, the amount of first cardiovascular events (7,3%) in our population is similar to other studies (Holtermann, 2021), but the few events may have limited our statistical power, yet the long follow-up time (15 years) is a strength. Consequently, we have less power in the models with the most parameters (e.g. Model C, Model IV), explaining the wide confidence intervals associated with their point estimates. Further studies using device-based assessment of physical activity are needed to confirm our findings.

    We have now extended our discussion on power. See line 297-300.

Even though the ABC cohort was a nationally representative sample, as in any research study, the participants may be healthier and more physically active than the general population which may affect the number of events. Still, the amount of first cardiovascular events (7,3%) in our population is similar to other studies [13], but the sample size and number of events may have limited our statistical power. Consequently, we have less power in the models with the most parameters (e.g. Model C, Model IV), explaining the wide confidence intervals associated with their point estimates. Future studies using device-based assessment of physical activity are needed to confirm our findings.

  1. Regarding domains for physical activity:
    We agree that we should elaborate more on this in the discussion. We found an effect of total time in MVPA, when split by domain we find a similar trend but not significant. This can be a power problem as well as highlighting how difficult it is to measure MVPA using accelerometry in different domains. The accelerometer measured MVPA, i.e using the same measure for MVPA in both domains seem to yield the same positive effect. Future research should also investigate the possible differences between measured and perceived intensity and amount of physical activity performed at work, since it is suggested that many occupations may not be as active as reported. Even though we used both self-reports and movement devices, our study was not primary designed to assess these associations.

We have elaborated more on this in the discussion see line 300-307.

To distinguish the domains in which the activity occurs, we recommend that future studies will use a complementary diary that could enable a cross-validation of the data sources. Future research should also investigate the possible differences between measured and perceived intensity and amount of physical activity performed at work, since it is suggested that many occupations may not be as active as reported [29]. Even though we used both self-reports and movement devices, we are aware that these two methods assess physical activity in different ways, associated with various limitations, that affect our findings. Last, we consider it important to mention that the present study is observational, and reverse causation is an issue that remains unknown.